# Electrolyte engineering via ether solvent fluorination for developing stable non-aqueous lithium metal batteries

Yan Zhao[1,4], Tianhong Zhou[1,4], Mounir Mensi[2], Jang Wook Choi ⬡[3] ✉ & Ali Coskun ⬡[1] ✉

Fluorination of ether solvents is an effective strategy to improve the electrochemical stability of non-aqueous electrolyte solutions in lithium metal batteries. However, excessive fluorination detrimentally impacts the ionic conductivity of the electrolyte, thus limiting the battery performance. Here, to maximize the electrolyte ionic conductivity and electrochemical stability, we introduce the targeted trifluoromethylation of 1,2-dimethoxyethane to produce 1,1,1-trifluoro-2,3-dimethoxypropane (TFDMP). TFDMP is used as a solvent to prepare a 2 M non-aqueous electrolyte solution comprising bis(fluorosulfonyl)imide salt. This electrolyte solution shows an ionic conductivity of 7.4 mS cm$^{-1}$ at 25 °C, an oxidation stability up to 4.8 V and an efficient suppression of Al corrosion. When tested in a coin cell configuration at 25 °C using a 20 µm Li metal negative electrode, a high mass loading LiNi$_{0.8}$Co$_{0.1}$Mn$_{0.1}$O$_2$-based positive electrode (20 mg cm$^{-2}$) with a negative/positive (N/P) capacity ratio of 1, discharge capacity retentions (calculated excluding the initial formation cycles) of 81% after 200 cycles at 0.1 A g$^{-1}$ and 88% after 142 cycles at 0.2 A g$^{-1}$ are achieved.

In the pursuit of battery technologies with higher energy densities, lithium (Li) metal batteries (LMBs) using metallic Li (theoretical specific capacity of 3860 mAh g$^{-1}$) to replace the traditional graphite (theoretical specific capacity of 372 mAh g$^{-1}$) anode have received significant attention in recent years[1–3]. However, the poor cycle life of LMBs limits their practical application, which mainly arises from severe side reactions between Li metal anode (LMA) and non-aqueous electrolyte solution, mechanically unstable solid electrolyte interphase (SEI) formation, low Columbic efficiency (CE), Li dendrite growth and formation of 'dead' Li (i.e., Li metal regions which are electronically disconnected from the current collector) during battery cycling[4–6]. Moreover, these detrimental phenomena are intertwined to one another, making it more difficult to fundamentally resolve the problem.

Electrolyte engineering is a promising approach owing to its easy adaptability to the existing battery manufacturing scheme. Conventional non-aqueous electrolyte solutions involving carbonate and ether based solvents exhibit either low thermodynamic stability towards Li metal causing severe parasitic reactions or poor anodic stability (<4 V) with limited electrochemical window at moderate salt concentrations (i.e., 1 M salt), respectively[7,8]. Several strategies targeting the optimization of electrolyte formulations have been introduced to stabilize the SEI on LMA and expand the oxidization window to match the competitive layer-structured cathode; these approaches are high concentration electrolytes (HCEs)[9–11], localized high concentration electrolytes (LHCEs)[12–15], electrolyte additives[16–18], fluorinated amide[19], mixed salt electrolytes[20,21], and weakly solvating solvents[10,22–26]. Recently,

[1]Department of Chemistry, University of Fribourg, Fribourg 1700, Switzerland. [2]Institute of Chemical Sciences and Engineering (ISIC), École Polytechnique Fédérale de Lausanne, Sion 1950, Switzerland. [3]School of Chemical and Biological Engineering, Department of materials science and engineering, and Institute of Chemical Processes, Seoul National University, 1 Gwanak-ro, Gwanak-gu, Seoul 08826, Republic of Korea. [4]These authors contributed equally: Yan Zhao, Tianhong Zhou. ✉e-mail: jangwookchoi@snu.ac.kr; ali.coskun@unifr.ch

cyclic diluent was introduced to optimize solvation structure, stabilize both anode and cathode interphase and improve cycling performance at low rates[27]. In these examples, common strategy is to reduce the amount of free solvent molecules to alleviate their side reactions on both anode and cathode surfaces, and promote anion decomposition by forming strong Li⁺-anion coordination clusters to achieve robust and inorganic-rich SEI[28,29]. In this direction, several fluorinated ether-based solvents with moderate salt concentrations have shown promising LMA performance to enable anion-derived inorganic SEI, arising from the decreased solvation power of oxygen atoms after fluorination and/or steric effects[30,31]. These solvents also showed improved compatibility towards high voltage nickel (Ni)-rich cathodes. 1,2-dimethoxyethane (DME) has been extensively used as electrolyte solvent owing to its low viscosity and strong solvation power, which promote high ionic conductivity[9]. On the other hand, the low anodic stability of DME restricts its application when paired with high voltage Ni-rich cathodes. Recently, Liu et al. reported anodic decomposition of DME was suppressed by electric double layer induced from LiNO₃ additive[8]. The decomposition of DME is mainly derived from the H-abstraction[32,33] from central ethyl group promoting the formation of unsaturated intermediates. Moreover, the oxidation stability of DME also involves the removal of lone pair electrons of the oxygen atom[32,34]. 1,2-dimethoxypropane (DMP) was introduced to decrease the solvation power through steric effect to promote the anion−Li⁺ interactions while providing a modest increase in the oxidation stability compared to DME[35]. Whereas substituting an ethyl proton with an electron-withdrawing group such as fluorine and decreasing the highest occupied molecular orbital (HOMO) energy level could increase the oxidation stability, high degree of fluorination decreases the solvation power, salt solubility and ionic conductivity, thus hindering the operation of LMBs during their repeated charge-discharge cycles. Therefore, it is pivotal to balance the ionic conductivity and high oxidization stability through rational structural control and targeted functionalization of solvent molecule.

In this work, in order to realize an anti-oxidation property, good solvation ability and high ionic conductivity simultaneously, we substituted the reactive proton on the ethyl moiety with a -CF₃ group to form 1,1,1-trifluoro-2,3-dimethoxypropane (TFDMP). A comparison with the -CH₃ substituted version, DMP, allowed us to probe the impact of steric and electronic effects. When 2 M lithium bis(fluorosulfonyl)imide (LiFSI) salt was dissolved, TFDMP-based electrolyte showed improved physicochemical properties, which was manifested by its good oxidation stability up to 4.8 V, high ionic conductivity of 7.4 mS cm⁻¹ at 25 °C and suppressed Al corrosion. While Li∥Cu asymmetric coin cell with TFDMP-based electrolyte exhibited an average CE of 99.6%, Li∥Li symmetrical cell showed stable interface up to 1600 h at 1 mA cm⁻² with a cutoff capacity of 1 mAh cm⁻². Moreover, the assembly and testing of coin cells with Li metal (20 μm thick) as the negative electrode and LiNi₀.₈Co₀.₁Mn₀.₁O₂ (NCM811)-based positive electrode enable a discharge capacity retention (calculated excluding the initial formation cycle) of almost 100% after 450 cycles at 0.2 A g⁻¹ (1.6 mA cm⁻²) and also performed well at high rates up to 2 A g⁻¹ (16 mA cm⁻²). Mass of the specific current and specific capacity refers to the mass of the active material in the positive electrode. Assembly and testing of coin cells applying relevant practical conditions such as a 20 μm Li metal negative electrode, a 20 mg cm⁻² NCM811-based positive electrode with an N/P (negative/positive capacity) ratio of 1.0 achieved 81% discharge capacity retention (calculated excluding the initial formation cycles) after 200 cycles at 0.1 A g⁻¹ (2 mA cm⁻²) and 88% discharge capacity retention (calculated excluding the initial formation cycles) after 142 cycles at 0.2 A g⁻¹ (4 mA cm⁻²).

## Results

### Design, characterization, and evaluation of electrolytes

Although fluorinated ether solvent-based electrolyte solutions exhibit promising electrochemical performance owing to their modified solvation structure involving the formation of contact ion pairs (CIP) and aggregate (AGG) species in Li metal cells[31], the excessively strong Li⁺-anion coordination hinders the Li⁺ diffusion and in turn compromises the ionic conductivity and cell impedance[23]. In order to address this challenge, a targeted functionalization strategy of DME was developed to alleviate its decomposition pathways towards improving oxidation stability while retaining high ionic conductivity. The decomposition of DME involves the abstraction of ethyl proton and a subsequent elimination step as evidenced by the formation of unsaturated species after cycling[32,33]. Moreover, the oxidation of DME relates to lone pair electrons on oxygen atoms[34]. In the presence of Li ions, however, DME forms a five-membered ring chelation structure with Li⁺[22], which significantly lowers its HOMO energy level and improves its oxidative stability, as verified by the high oxidative stability of HCEs based on DME. In order to tackle this problem and realize high oxidative stability at relatively low salt concentrations (i.e., 2 M salt)[36], we targeted both steric and electronic control by replacing the one of the ethyl protons with a trifluoromethyl group, which would alleviate the H-abstraction, while simultaneously decreasing the solvation power of O atoms. Accordingly, we also prepared the methyl substituted version to probe the steric effects[10] and clearly understand the role of electronic effects. Whereas direct F-substitution of the central alkyl moiety in ether solvents has markedly improved the oxidation stability, it results in a low ionic conductivity (3.5 mS cm⁻¹ at an undisclosed testing temperature)[22]. In this sense, the introduction of trifluoromethyl group would help to achieve both good oxidation stability and high ionic conductivity owing to its proximity to the O atoms (Fig. 1a). While DMP solvent is commercially available, TFDMP was synthesized through a single-step methylation of 1,1,1-trifluoro-2,3-propanediol in 41% yield. The structure of TFDMP was verified by nuclear magnetic resonance (NMR) spectroscopy analysis (Supplementary Fig. 1–3). The calculations of the HOMO and the lowest unoccupied molecular orbital (LUMO) energy values were performed to compare the effect of functional groups (Supplementary Fig. 4). Compared with DME (−7.19 eV), DMP exhibited a slightly higher HOMO value of −7.14 eV originating from the weakly electron donating methyl group, whereas TFDMP showed the lowest HOMO value of −7.58 eV owing to the presence of the strongly electron withdrawing trifluoromethyl group. LUMO values on the other hand decreased going from DME to DMP and to TFDMP. Based on our previous study[31], 2 M LiFSI salt was dissolved in the three solvents to prepare the electrolytes (referred to as 2 M LiFSI-DME, 2 M LiFSI-DMP and 2 M LiFSI-TFDMP). The effect of different salts and salt concentrations in electrolytes was also compared (Supplementary Fig. 5). Electrolytes using LiFSI displayed better CE performance than using lithium hexafluorophosphate (LiPF₆) and lithium bis(trifluoromethanesulfonyl)imide (LiTFSI), salt concentration of 2 M and 3 M exhibited similar and better CE stability than 1 M, but higher salt concentration increases the cost and viscosity[36], accordingly, we used the 2 M LiFSI in all electrolytes. First, linear sweep voltammetry (LSV) measurements were conducted by using Li∥Al asymmetric Swagelok cells at a scan rate of 0.5 mV s⁻¹ (Fig. 1b). 2 M LiFSI-DME exhibited a decomposition at around 4 V, while 2 M LiFSI-DMP showed improved anti-oxidation up to 4.4 V, which verified the positive role of steric effect, even with a higher HOMO value of DMP compared to that of DME. 2 M LiFSI-TFDMP showed an oxidation stability up to 4.8 V originating from the combination of steric and electronic effects owing to the incorporation of -CF₃ moiety. Severe corrosion of Al current collector has been reported for the electrolytes containing FSI anions[37,38]. Accordingly, we performed Al corrosion experiments of the three electrolytes in Li∥Al asymmetric coin cells by holding a constant-voltage at 5 V for 24 hours (Supplementary Fig. 6).

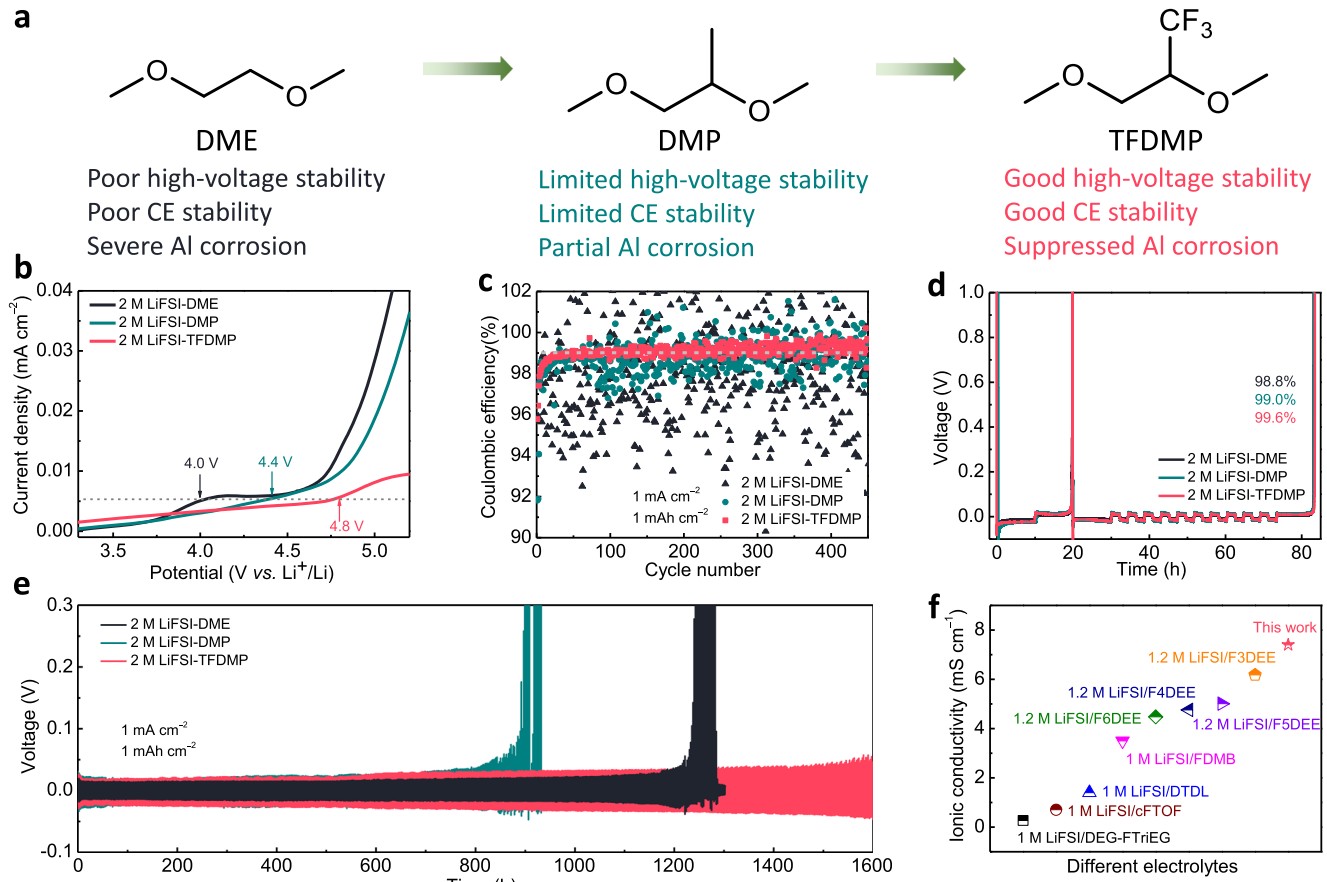

**Fig. 1 | Schematic of solvent fluorination and electrochemical characterizations of various non-aqueous electrolyte solutions. a** Design of solvent structure and associated properties. **b** Oxidation stability of electrolytes in Li‖Al asymmetric cells tested by linear sweep voltammetry (LSV) at a scan rate of 0.5 mV s$^{-1}$, the gray dotted line represents the same oxidation current density of three electrolytes. **c** CE test of Li‖Cu asymmetric cells using different electrolytes at 1 mA cm$^{-2}$ with a cutoff capacity of 1 mAh cm$^{-2}$. **d** Average CE test of three electrolytes by Aurbach method[40]. **e** Voltage−time profiles of Li‖Li symmetric cells with different electrolytes. Testing temperature is 25 ± 1 °C. **f** Comparisons of bulk ionic conductivities with previously reported electrolytes based on fluorinated-ether solvents[22,23,31,44,45], reference articles cited did not report the testing temperature for the ionic conductivity.

The Al foil with 2 M LiFSI-DME showed significant cracks and delamination. The Al foil with 2 M LiFSI-DMP on the other hand displayed partial cracks. In contrast, 2 M LiFSI-TFDMP electrolyte realized flat and smooth Al foil. Cyclic voltammetry (CV) of Li‖Cu asymmetric coin cells with three electrolytes was performed to probe their compatibility toward Li anode (Supplementary Fig. 7). Compared to the other two electrolytes, 2 M LiFSI-TFDMP exhibited highly overlapping peaks verifying negligible side reactions and stable interface formation. Long-term cycling CE tests using Li‖Cu asymmetric coin cells were also conducted at 1 mA cm$^{-2}$ with a cutoff capacity of 1 mAh cm$^{-2}$ with different electrolytes to quantitatively investigate the reversibility of Li plating and stripping (Fig. 1c). The cell with 2 M LiFSI-DME exhibited massive fluctuations and an average CE of only 98.2% within 450 cycles owing to the unstable SEI and dead Li formation. The cell with 2 M LiFSI-DMP displayed smaller fluctuation and higher average CE of 98.8%. In comparison, 2 M LiFSI-TFDMP electrolyte showed stable and the highest average CE of 99.1%. The corresponding charge-discharge profile of Li‖Cu asymmetric coin cell with DME-based electrolyte revealed the origin of its low CE as irreversible Li plating (Supplementary Fig. 8). In addition, CE tests at 3 mA cm$^{-2}$ with a cutoff capacity of 3 mAh cm$^{-2}$ using different electrolytes were also shown in Supplementary Fig. 9. The performance of the cell with 2 M LiFSI-TFDMP electrolyte was found to be improved compared to other electrolytes and showed a stable CE above 99% under these conditions. Electrochemical impedance spectroscopy (EIS) measurements and analyses

of Li‖Cu asymmetric coin cells with different electrolytes were also performed to investigate anode-electrolyte interfacial stability (Supplementary Fig. 10 and Supplementary Table 1). The cell with 2 M LiFSI-DME showed an increase in the charge transfer resistance after only 20 cycles indicating sluggish Li$^+$ ion migration from the accumulated SEI[39]. Furthermore, average CE test by modified Aurbach method[40] (plating 5 mAh cm$^{-2}$ Li on Cu substrate then cycling 1 mAh cm$^{-2}$ Li for 10 cycles and calculating the CE after fully stripping until 1 V) was performed to avoid side reaction with the Cu substrate (Fig. 1d). The CE of cells using 2 M LiFSI-DME, 2 M LiFSI-DMP and 2 M LiFSI-TFDMP electrolytes were found to be 98.8%, 99.0%, and 99.6% (average CEs based on 10 cycles), respectively. The difference in the CE values of these electrolytes points to the impact of molecular structure to tune the solvation structure and to facilitate stable SEI formation. Moreover, Li‖Li symmetrical coin cells with different electrolytes were also evaluated at 1 mA cm$^{-2}$ with a cutoff capacity of 1 mAh cm$^{-2}$ to investigate the interfacial stability and overpotential (Fig. 1e). The cell with 2 M LiFSI-DME exhibited a stable interface for 1200 h followed by a severe increase in the polarization. 2 M LiFSI-DMP electrolyte on the other hand showed only 800 h stability resulting from the side reaction between DMP and Li metal presumably owing to the integration of weakly electron donating -CH$_3$ group compared to DME, in agreement with the CV tests (Supplementary Fig. 7). In stark contrast, symmetrical cell with 2 M LiFSI-TFDMP electrolyte maintained a stable interface for 1600 h owing to robust SEI formation on the Li metal surface. These

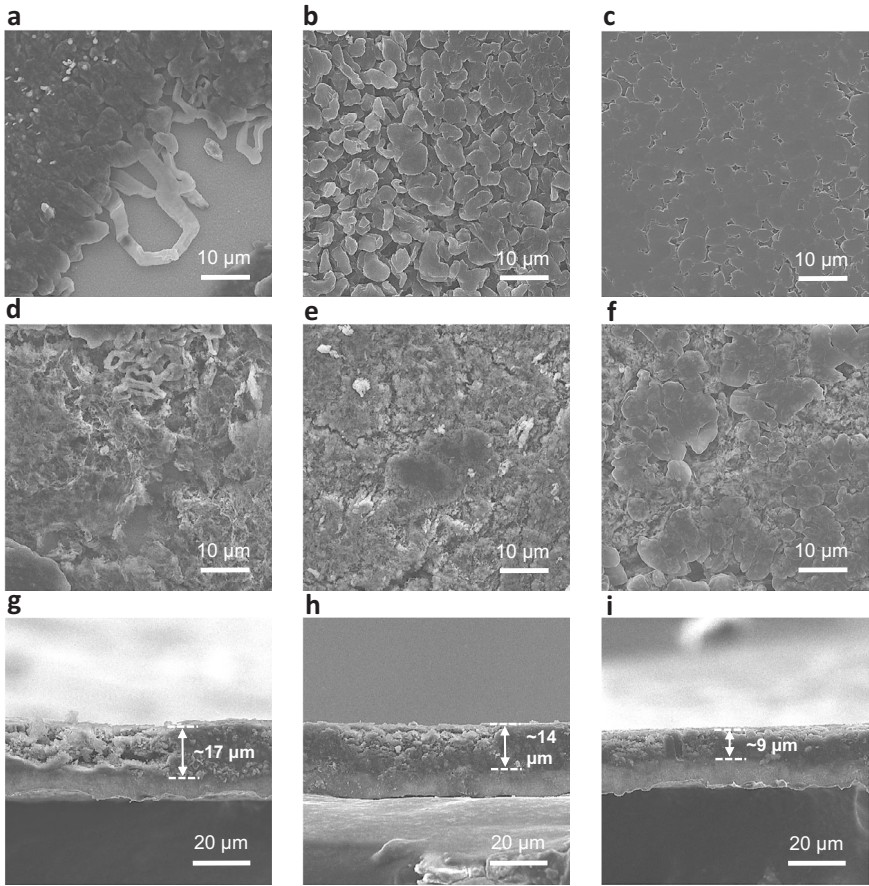

**Fig. 2 | Ex situ postmortem microscopy measurements of electrodes cycled in Li‖Cu cell configuration. a–c** SEM images of Li plating on Cu foil after the first cycle at 1 mA cm$^{-2}$ with a cutoff capacity of 2 mAh cm$^{-2}$ in 2 M LiFSI-DME (**a**), 2 M LiFSI-DMP (**b**), and 2 M LiFSI-TFDMP (**c**). **d–i** Top-view and cross-sectional view SEM images of Li plating on Cu foil after 20 cycles at 1 mA cm$^{-2}$ with a cutoff capacity of 1 mAh cm$^{-2}$ in 2 M LiFSI-DME (**d, g**), 2 M LiFSI-DMP (**e, h**), and 2 M LiFSI-TFDMP (**f, i**). The thicknesses in **g–i** represented plated Li.

results point to the critical role of electronic effects and limited impact of steric effects for Li metal interfacial stability with the solvents. As fluorinated ether solvents often sacrifice their intrinsic ionic conductivities,[22] the ionic conductivities of three electrolytes were measured by symmetrical coin cells with stainless steel (SS) blocking electrodes. 2 M LiFSI-DME, 2 M LiFSI-DMP and 2 M LiFSI-TFDMP electrolytes showed bulk ionic conductivity values of 18.1, 14.0 and 7.4 mS cm$^{-1}$ at 25 °C (Supplementary Fig. 11), respectively. Importantly, 2 M LiFSI-TFDMP electrolyte showed the highest ionic conductivity among the reported electrolytes based on fluorinated ether solvents (Fig. 1f). The lithium ion transference numbers ($t_{Li^+}$) of 2 M LiFSI-DME, 2 M LiFSI-DMP and 2 M LiFSI-TFDMP were measured by using the same Li‖Li symmetric coin cells and the corresponding values were 0.38, 0.55 and 0.60 (Supplementary Fig. 12), respectively. The highest $t_{Li^+}$ of 2 M LiFSI-TFDMP originated from the decreased solvation power of TFDMP and stronger electrostatic interactions between Li$^+$ and FSI anion compared to the other tested electrolytes. The physicochemical properties of solvents and electrolytes are summarized in the Supplementary Table 2. Safety performance and thermal stability of each electrolyte were also evaluated by self-extinguishing time (SET) and thermogravimetric analysis (TGA) (Supplementary Table 3 and Supplementary Fig. 13).

## Li plating morphology and evolution

Li plating morphologies on Cu substrate with different electrolytes were investigated by using Li‖Cu asymmetric coin cells. Ex situ postmortem scanning electron microscopy (SEM) measurements and analyses were carried out to understand the influence of solvent on the Li plating morphology (Fig. 2 and Supplementary Fig. 14). First, 2 mAh cm$^{-2}$ Li was plated on Cu foil at 1 mA cm$^{-2}$ current density (Fig. 2a–c), the plated Li in the presence of 2 M LiFSI-DME electrolyte showed an uneven coverage and bare Cu substrate along with Li whiskers, 2 M LiFSI-DMP electrolyte on the other hand realized uniform deposition, random Li grains and several voids. Plated Li from 2 M LiFSI-TFDMP presented flat, tightly packed, large grains, thus alleviating the side reactions originating from overexposure of metallic Li to the electrolyte and decreasing the formation of 'dead' Li during stripping. Specifically, this preferred morphology guaranteed highly reversible Li plating and stripping, which enabled stable and high CE values in Fig. 1c, d. In order to observe the evolution of plated Li morphology, Li‖Cu asymmetric cells were cycled for 20 cycles. The plated Li exhibited dendrite-like morphology depositions and accumulated SEI with 2 M LiFSI-DME (Fig. 2d) derived from the unstable SEI and 'dead' Li formation during cycling, showed thick SEI layer with 2 M LiFSI-DMP (Fig. 2e) and obtained uniform Li grains with 2 M LiFSI-TFDMP (Fig. 2f) owing to the robust SEI formation, the related thicknesses of plated Li from cross-sectional view (Fig. 2g–i) were 17 μm, 14 μm, and 9 μm, respectively. These results clearly demonstrate the influence of solvent structure on the Li plating morphology and its evolution during cycling.

## Analysis of solvation structure

In order to understand the improved LMA performance and high ionic conductivity of TFDMP solvent, solvation structures of three electrolytes were comparatively analyzed by theoretical and experimental characterizations (Fig. 3). Electrostatic potential maps (ESP) were

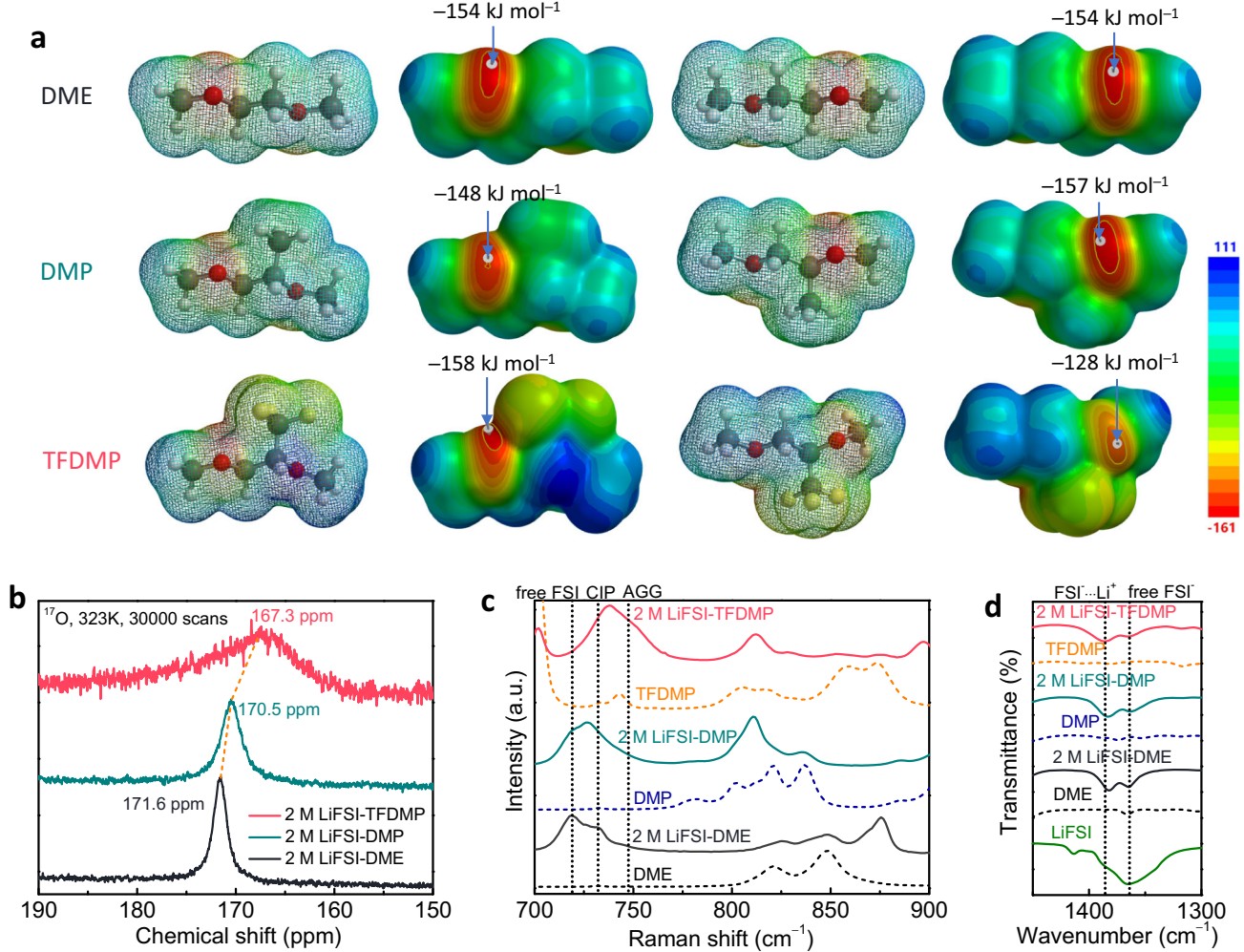

**Fig. 3 | Solvents and electrolyte solutions physicochemical characterizations.** **a** Electrostatic potential (ESP) maps of DME, DMP and TFDMP solvents with front and back views, color scheme: yellow, F; red, O; gray, C; white, H. **b** $^{17}$O NMR spectra (323 K) of different electrolytes using 1 M LiCl-D$_2$O as a standard (0 ppm) for calibration. **c** Raman spectra of different solvents and electrolytes obtained using 785 nm laser at 25 ± 1 °C. **d** FTIR spectra of different solvents and electrolytes at 25 ± 1 °C.

calculated to compare the influence of functional groups on the electron density of ether oxygens (Fig. 3a). Minimal value on oxygen in DME was found to be −154 kJ mol$^{-1}$, after methylation, however, the value on the oxygen atom adjacent to -CH$_3$ group in DMP decreased to −157 kJ mol$^{-1}$, implying increased electron density. In the case of TFDMP, the value on oxygen close to -CF$_3$ group increased to −128 kJ mol$^{-1}$, indicating significantly lower electron density compared to both DME and DMP, thus leading to a decreased Li$^+$ solvation power. Moreover, $^{17}$O NMR of electrolytes was recorded to compare their solvation structures (Fig. 3b). Going from 2 M LiFSI-DME to 2 M LiFSI-DMP and to 2 M LiFSI-TFDMP, the chemical shift of sulfonyl oxygen in the FSI anion showed an upfield shift from 171.6 to 170.5 then to 167.3 ppm, respectively, revealing increased FSI$^-$ and Li$^+$ interactions[14] through solvent structure control. Raman spectroscopy is widely adopted to analyze the nature of solvation clusters in electrolytes, such as free anions, CIPs and AGGs[41,42]. As shown in Fig. 3c, there were significant amounts of free FSI anions (719 cm$^{-1}$) and some CIPs (732 cm$^{-1}$) in the 2 M LiFSI-DME electrolyte. The signal intensity of CIPs clusters was found to be the same as free FSI anion in 2 M LiFSI-DMP owing to the decreased solvation power of DMP through steric effects, which limit the interaction of the solvent with Li$^+$. In contrast, 2 M LiFSI-TFDMP mainly showed high amounts of CIPs and AGGs (747 cm$^{-1}$). These results are in line with the bulk ionic conductivity data obtained for these electrolytes (Supplementary Table 2), that is stronger

interactions between Li$^+$ and FSI anion leads to a lower ionic conductivity. These results also verify the critical role of targeted functionalization approach to maintain high ionic conductivity as the overfluorinated solvents/diluents exhibit significantly lower ionic conductivities. Furthermore, Fourier-transform infrared (FTIR) spectroscopy measurements on different electrolytes was also employed to probe the solvation structure (Fig. 3d and Supplementary Fig. 15). The peaks at 1385 cm$^{-1}$ and 1364 cm$^{-1}$ were assigned to the Li$^+$-FSI$^-$ interaction and free FSI$^-$, respectively. Going from 2 M LiFSI-DME to 2 M LiFSI-DMP and then to 2 M LiFSI-TFDMP, the peaks at 1385 and 1364 cm$^{-1}$ became broader and weaker, indicating the increased coordination of Li$^+$-FSI$^-$ and decreased free FSI$^-$[22]. NMR, Raman and FTIR results coherently reflected impact of steric and electronic effects on the solvation structure of electrolytes.

## Electrochemical energy storage behavior in Li||LiNi$_{0.8}$Co$_{0.1}$Mn$_{0.1}$O$_2$ cell configuration

Based on the Li anode compatibility and high-voltage endurance of TFDMP-based electrolyte, Li metal electrodes paired with NCM811-based positive electrodes were further evaluated using coin cells (Fig. 4), an additional piece of blank Al current collector (14 μm) and Al-coated (5 μm thick) stainless steel cathode case were used to suppress electrolyte corrosion on stainless steel cathode case[8,14]. Formation cycle at low specific current (0.02 A g$^{-1}$) was adopted to form stable

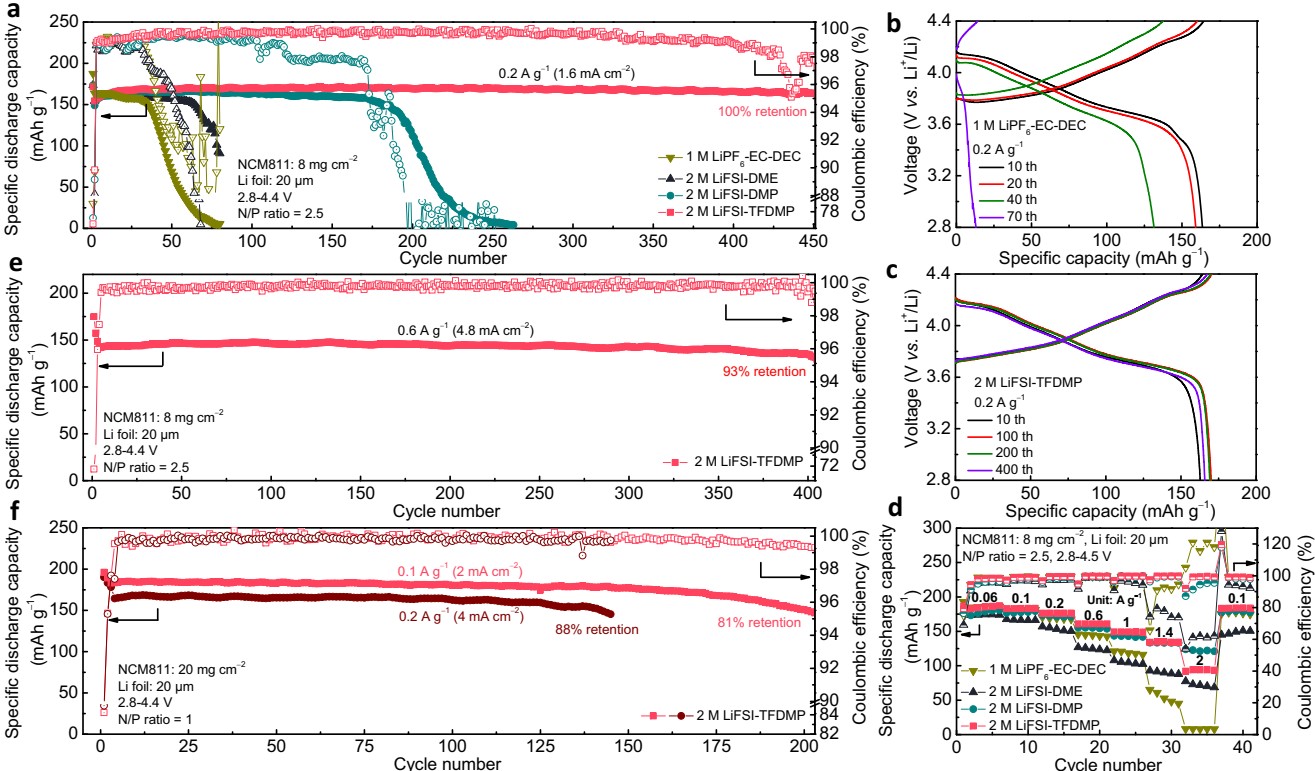

**Fig. 4 | Electrochemical energy storage performance of Li‖NCM811 cells with various electrolyte solutions. a** Cycling test of Li‖NCM811 cells with different electrolytes at 0.2 A g⁻¹ (1.6 mA cm⁻²) after one formation cycle at 0.02 A g⁻¹. **b, c** The corresponding charge-discharge profiles at different cycles with 1 M LiPF₆-EC-DEC (**b**) and 2 M LiFSI-TFDMP (**c**). **d** Rate performance of Li‖NCM811 cells with different electrolytes after one formation cycle at 0.02 A g⁻¹ (keeping charge and discharge at the same specific current). **e** Cycling test of Li‖NCM811 cells with 2 M

LiFSI-TFDMP electrolyte at 0.6 A g⁻¹ (4.8 mA cm⁻²) after formation cycles of 0.02 A g⁻¹, 0.2 A g⁻¹, and 0.4 A g⁻¹, one cycle for each. **f** Performance of high loading Li‖NCM811 cells with 2 M LiFSI-TFDMP electrolyte at 0.1 A g⁻¹ (2 mA cm⁻²) and 0.2 A g⁻¹ (4 mA cm⁻²) after formation cycles of 0.02 A g⁻¹, 0.06 A g⁻¹, and 0.1 A g⁻¹, one cycle for each. The mass of the specific current and specific capacity referred to the mass of the active material in the positive electrode. All tests were conducted in coin cells at 25 ± 1 °C.

interphases on both anode and cathode. Under moderate conditions, that are Li foil (20 μm) paired with NCM811 (8 mg cm⁻²) with the N/P ratio of 2.5 at 0.2 A g⁻¹ (1.6 mA cm⁻²), the cell with carbonate electrolyte of 1 M LiPF₆-EC-DEC lasted for 35 cycles followed by a drastic decay. The cell using 2 M LiFSI-DME maintained a stable capacity for only 60 cycles owing to the continuous decomposition of DME solvent[22]. The cell with 2 M LiFSI-DMP stabilized for 175 cycles, which was followed by a fast capacity decay derived from the continuous consumption of metallic Li[35]. In stark contrast, the cell with 2 M LiFSI-TFDMP realized 100% capacity retention (from 2nd cycle to 451st cycle) after 450 cycles (Fig. 4a), which proved the viability of TFDMP solvent in a practical cell setting. The corresponding charge-discharge profiles in Fig. 4b, c and Supplementary Fig. 16 revealed the irreversible capacity when the cells failed. The rate performance of cells using different electrolytes was tested from 0.06 A g⁻¹ to 2 A g⁻¹ with the charge and discharge specific current fixed equal (Fig. 4d), the related zoom-in CEs were shown in Supplementary Fig. 17. The cell with 1 M LiPF₆-EC-DEC showed fast decay beyond 1 A g⁻¹, and only maintained about 48 mAh g⁻¹ capacity (25% retention, from 1st cycle to 30th cycle) and 91.7% CE at 1.4 A g⁻¹. The cell with 2 M LiFSI-DME started to become unstable at 1 A g⁻¹ with 96.0% CE and showed about 89 mAh g⁻¹ capacity (50% retention, from 1st cycle to 30th cycle) and 73.9% CE at 1.4 A g⁻¹. In contrast, the cells with 2 M LiFSI-DMP and 2 M LiFSI-TFDMP at 1.4 A g⁻¹ exhibited a similar capacity of 134 mAh g⁻¹ (76% retention, from 1st cycle to 30th cycle) and CE above 99.5%. When increasing to 2 A g⁻¹ (16 mA cm⁻²), the cell with 2 M LiFSI-TFDMP showed much higher CE of 99.4% than 95.0% of 2 M LiFSI-DMP. These results reflected the advantage of high ionic conductivity of 2 M LiFSI-TFDMP and its Li anode stability to achieve good rate performance. Considering the ionic conductivity of TFDMP-

based electrolyte compared to those of other reported electrolytes (Fig. 1f), Li metal cell was further tested by long cycling at 0.6 A g⁻¹ (4.8 mA cm⁻²), which exhibited 93% retention (from 4th cycle to 403rd cycle) after 400 cycles (Fig. 4e). In an effort to benchmark practical cell conditions, a Li foil (20 μm) was paired with a high-loading NCM811 laminate (20 mg cm⁻²) by the N/P ratio of 1.0 (Fig. 4f), the corresponding cell with 2 M LiFSI-TFDMP still showed a good long-term cycling performance of 81% retention (from 3rd cycle to 202nd cycle) after 200 cycles at 0.1 A g⁻¹ (2 mA cm⁻²) and 88% retention (from 4th cycle to 145th cycle) after 142 cycles at 0.2 A g⁻¹ (4 mA cm⁻²). Furthermore, the cell using same configuration was cycled under a lean electrolyte (6 g Ah⁻¹) condition (Supplementary Fig. 18) and it still realized 93% retention (from 4th cycle to 103rd cycle) after 100 cycles at 0.2 A g⁻¹ (4 mA cm⁻²). Performance comparison of cells with different electrolytes was summarized in Supplementary Table 4.

## Ex situ postmortem physicochemical characterizations of Li‖NCM811 electrodes

As the cell performance is directly related to the interfacial characteristics through electrolyte decomposition, X-ray photoelectron spectroscopy (XPS) analysis was conducted to compare the SEI and cathode electrolyte interphase (CEI) compositions with different electrolytes in Li‖NCM811 cells after cycling (Fig. 5 and Supplementary Figs. 19–23). Depth profiles in Fig. 5a, d revealed the distributions of elements in the SEI layers on Li anodes along the depth. Compared to the carbonate electrolyte of 1 M LiPF₆-EC-DEC, ether-based electrolytes exhibited more uniform distributions of elements along the depth. The SEI layer on Li anode with 2 M LiFSI-TFDMP contained higher F (15.6%) and S (9.2%) contents after 12 minutes sputtering compared to the F

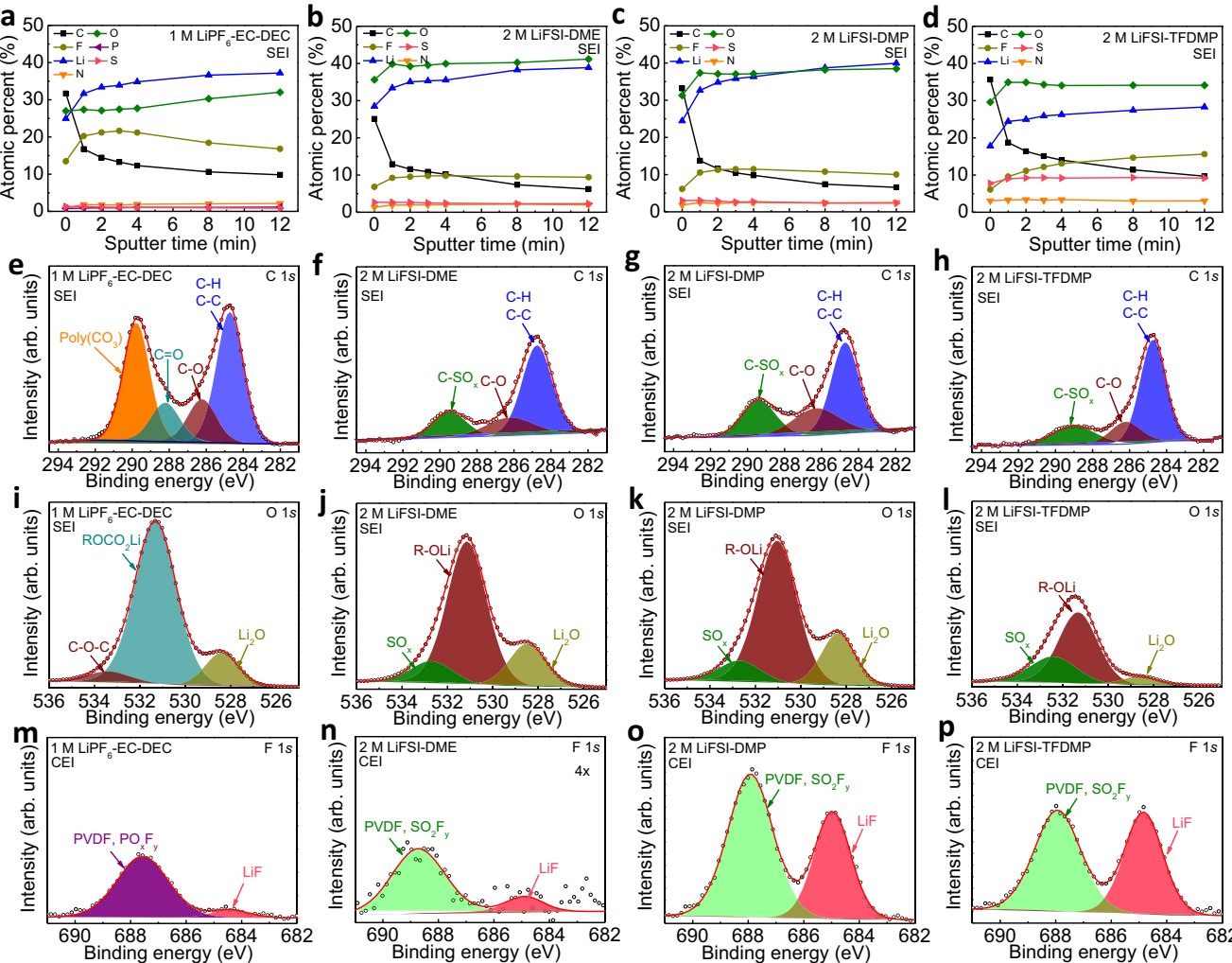

**Fig. 5 | Ex situ postmortem XPS characterization of negative and positive electrodes harvested from Li‖NCM811 cells. a–d** Depth profiles of SEI layers on Li anode surface in Li‖NCM811 cells after cycling at 0.2 A g⁻¹ for 20 cycles with 1 M LiPF₆-EC-DEC (**a**), 2 M LiFSI-DME (**b**), 2 M LiFSI-DMP (**c**), and 2 M LiFSI-TFDMP (**d**). **e–l** C 1*s* and O 1*s* XPS profiles of SEI layers on Li anodes after 12 minutes sputtering with 1 M LiPF₆-EC-DEC (**e, i**), 2 M LiFSI-DME (**f, j**), 2 M LiFSI-DMP (**g, k**) and 2 M LiFSI-TFDMP (**h, l**). **m–p** F 1*s* XPS profiles of CEI layers on NCM811 cathodes with 1 M LiPF₆-EC-DEC (**m**), 2 M LiFSI-DME (**n**), 2 M LiFSI-DMP (**o**), and 2 M LiFSI-TFDMP (**p**). Electrodes were harvested at a fully discharged (2.8 V) cell state.

(10.0%) and S (2.3%) contents with 2 M LiFSI-DMP and F (9.4%) and S (2.3%) contents with 2 M LiFSI-DME, which indicated more FSI anion decomposition by the stronger coordination between cation and anion in 2 M LiFSI-TFDMP electrolyte, in agreement with the solvation structure in Fig. 3. From the C 1*s* profiles in Fig. 5e, h, solvent decomposition with 1 M LiPF₆-EC-DEC became evident from the strong poly(CO₃) peak at 289.9 eV. The area proportion of the C-O peak with respect to total C in ether-based electrolytes were found to be 19.3%, 24.7%, and 16.5% for the DME, DMP, and TFDMP-based electrolytes, respectively, which demonstrated that the solvent decomposition was hindered after trifluoromethylation. The O 1*s* profiles in Fig. 5i, l also exhibited the weakest R-OLi peak with 2 M LiFSI-TFDMP electrolyte. In Supplementary Fig. 21, the S 2*p* profiles showed higher amounts of SO₂F and SOₓ species with respect to total S in the SEI layer with 2 M LiFSI-TFDMP electrolyte compared to its DME- and DMP-based counterparts, thus pointing to the preferential decomposition of FSI. CEI layer on NCM811 cathode with each electrolyte was also compared as shown in Fig. 5m–p, Supplementary Figs. 22 and 23, and Supplementary Table 5. The LiF contents in CEI with 1 M LiPF₆-EC-DEC, 2 M LiFSI-DME, 2 M LiFSI-DMP and 2 M LiFSI-TFDMP electrolytes were 8.7%, 14.5%, 37.8% and 45.7%, respectively. The highest LiF content in the CEI of 2 M LiFSI-TFDMP was derived from the desirable FSI decomposition.

The intensity of C=O species in the XPS O 1*s* profiles of the CEI in ether-based electrolytes were found to decrease going from DME to DMP and to TFDMP in Supplementary Fig. 23, thus further verifying the higher oxidation stability of 2 M LiFSI-TFDMP. The XPS analysis of SEI and CEI revealed the strong coordination of cation and anion in the TFDMP-based electrolyte, which promoted preferential anion decomposition to form stable interphases. CEI on the cathode after cycling was also analyzed by transmission electron microscopy (TEM) as shown in Supplementary Fig. 24. Due to the severe electrolyte decomposition on the cathode surface during charging, 2 M LiFSI-DME electrolyte exhibited a thick and uneven CEI layer of 10–52 nm. CEI thickness of 15–35 nm was observed for 2 M LiFSI-DMP electrolyte. CEI formed by 2 M LiFSI-TFDMP on the other hand showed a more uniform and thinner layer of ~6–16 nm.

## Discussion

A fluorinated ether solvent was developed through targeted structural functionalization to realize both high ionic conductivity and anti-oxidation property. The precise introduction of functional group onto the ethyl backbone of ether solvent mitigated the oxidative decomposition, while largely maintaining fast-ion conduction simultaneously. By using this solvent, both asymmetric and symmetric cells realized

stable long-term cyclability with good rate performance. These findings highlight the importance of in-depth understanding of solvent decomposition pathways and the targeted functionalization strategy for the development of next generation electrolytes as a promising alternative to the commonly applied (over)fluorination approach.

## Methods

### Materials
Iodomethane (99%) was purchased from Thermo Scientific. Sodium hydride (60% in mineral oil) was purchased from Aldrich. Anhydrous DME (99.5%, water content of 23 ppm) and 1 M LiPF$_6$-EC-DEC (v: v = 1: 1, water content of 7 ppm) electrolyte were purchased from Sigma. Anhydrous THF (water content below 10 ppm) was obtained from a solvent drying system passed by alumina column to remove trace water. 1,2-dimethoxypropane (DMP, > 98%, water content of 6 ppm after molecular sieves drying) was purchased from Tokyo Chemical Industry (TCI). Battery grade LiFSI (99.9%, Canrd), Al foil (14 μm, 99.6%), Cu foil (9 μm, 99.99%), and NCM811 laminate (8 mg cm$^{-1}$ with the thickness of 32 μm and 20 mg cm$^{-1}$ with the thickness of 79 μm, loading is only based on the active material, cathodes contain 94.5% active material, 2.5% PVDF and 3% Super-P by weight) were purchased from Guangdong Canrd New Energy Technology Co.,Ltd. Li disc (Φ16 mm, 600 μm, 99.95%) and Li foil (20 μm, 99.95%) were purchased from China Energy Lithium Co., Ltd. All the chemicals and reagents used without further purification unless otherwise stated.

### Synthesis
TFDMP: To a 1 L round-bottom flask, 400 mL dry THF and 24 g of NaH (1 mol) were added and stirred for 20 minutes at 0 °C under Ar atmosphere. Then, 52 g 1,1,1-trifluoro-2,3-propanediol[31] (0.4 mol) was slowly added to this mixture by using a syringe pump. Afterwards, the suspension was further stirred at 0 °C for 1 h. Then 62.4 mL iodomethane (1 mol) was added dropwise to the suspension at 0 °C. The mixture was stirred for 2 h at room temperature (25 ± 1 °C) and then refluxed overnight. After the completion of the reaction, the mixture was filtered off using a filter funnel (Por. 3, 125 mL, pore size 16–40 μm). Liquid phase was subjected to a fractional distillation by using 60 cm fractionating column. The product was collected at 92 °C under atmospheric pressure. Yield: 41%, purity: 99.5%. $^1$H NMR (400 MHz, CDCl$_3$) $\delta_H$ = 3.76-3.68 (m, 1H), 3.66-3.62 (m, 1H), 3.61 (q, 3H, $J$ = 0.61 Hz), 3.56-3.52 (m, 1H), 3.41 (s, 3H) ppm; $^{19}$F NMR (376 MHz, CDCl$_3$) $\delta_F$ = −75.62 (d, $J$ = 6.81 Hz, 3 F) ppm; $^{13}$C NMR (100 MHz, CDCl$_3$) $\delta_C$ = 124.4 (q, $J$ = 283.90 Hz), 79.1 (q, $J$ = 29.34 Hz), 70.4 (q, $J$ = 2.20 Hz), 60.6 (s), 59.5 (s) ppm.

### Electrolyte preparation
DME, DMP and TFDMP were dried by using 4 Å molecular sieves before use. Electrolytes were prepared by dissolving 374 mg LiFSI in 1 mL of each solvent. All electrolytes were stirred for 4 h at room temperature (25 ± 1 °C) in an Ar-filled glovebox (O$_2$ < 0.5 ppm, H$_2$O < 0.5 ppm). All electrolytes contained water content below 10 ppm detected by Karl Fischer Titrator.

### Material characterizations
$^1$H, $^{19}$F and $^{13}$C NMR spectra were recorded on 400 MHz Bruker NMR spectrometer at ambient temperature (298 K) using CDCl$_3$ as a solvent. $^{17}$O NMR of electrolytes was recorded on 500 MHz Bruker NMR spectrometer at 323 K with 30,000 scans in a coaxial NMR tube using 1 M LiCl-D$_2$O as a standard (0 ppm) for calibration. Fourier transform infrared (FTIR) spectra of electrolytes and solvents were measured by using Bruker TENSOR II. XPS depth profiles and surface measurements were carried out on an Axis Supra (Kratos Analytical) using the monochromated Ka X-ray line of an Aluminum anode. The pass energy of the analyzer was set to 40 and 80 eV with step sizes of 0.15 and 0.2 eV for the surface measurements and depth profiles, respectively. The

samples were electrically insulated from the sample stage and charge compensation was used to limit charging effects, hence, the spectra were referenced at 284.8 eV using the aliphatic component of the C 1$s$ line. The sputtering was performed with an argon ion gun set at 1 keV, and rastered at 4×4 mm$^2$, yielding to a sputter rate of ~2.5 nm/min as calibrated on a Ta$_2$O$_5$/Ta calibration sample. The samples were sputtered 0 (surface), 1, 2, 3, 4, 8, and 12 min, giving the following estimated depth sequence: 0, 2.5, 5, 7.5, 10, 20, and 30 nm. Raman spectroscopy was obtained on a Renishaw inVia confocal Raman microscope with a 785 nm laser. TGA analysis of electrolytes was conducted using a TGA/DSC 3+ STAR$^e$ System (Mettler Toledo) under N$_2$ gas flow (5 mL/min) at the ramp rate of 10 °C/min from 25–600 °C using a 100 μL aluminum crucible with a predrilled lid. Deposition morphology of Li metal on Cu foil and Al corrosion were acquired by field emission scanning electron microscopy (FE-SEM, Tescan Mira3 LM FE). Transmission electron microscopy of CEI on cathode after cycling was conducted by FEI Tecnai spirit at 120 kV using carbon film-coated Cu grid. All electrode samples disassembled from coin cells were rinsed with anhydrous DME and transferred under Ar atmosphere. Orbital energy (HOMO and LUMO) and electrostatic potential (ESP) maps of solvents were calculated by Spartan 14 using density functional theory conducted at B3LYP/6-311 + +G**. Self-extinguishing time (SET) measurements were carried out by adding 0.5 g electrolytes on glass fiber filters (Φ20 mm) and recording the burning time after ignition.

### Electrochemical measurements
All electrochemical tests were conducted by either Swagelok-cell or 2032-type coin-cell with 40 μL electrolyte except for the lean condition using 20 μL electrolyte for high-loading Li‖NCM811 cell. LSV was conducted by using a Li‖Al asymmetric cell at a scan rate of 0.5 mV s$^{-1}$. Al corrosion was tested by using a Li‖Al asymmetric cell and holding constant-voltage at 5 V for 24 h. CV was conducted by a Li‖Cu asymmetric cell at a scan rate of 0.5 mV s$^{-1}$. Ionic conductivities of electrolytes were measured by EIS with 5 mV amplitude in the frequency range of 500 kHz to 100 mHz at 25 °C using stainless steel blocking cell. EIS was measured using a Li‖Cu asymmetric cell after holding open circuit voltage for 0.5 h at quasi-stationary potential with a disturbance amplitude (5 mV) in the frequency range of 100 kHz to 10 mHz using potentiostatic signal by six points per decade of frequency and EIS plots were fitted by ZView. Li$^+$ transference number was obtained by a Li‖Li symmetric cell under a polarization potential of 10 mV according to the reference[43]. Above-mentioned tests were conducted by VMP3 (Bio Logic Science Instruments) electrochemical station. CE measurement was conducted by a Li‖Cu asymmetric cell after 10 formation cycles in the voltage range of 0-1.5 V with a 50 μA current. Li was plated onto a Cu foil at a steady current density, then stripped at same current density by the cut-off voltage of 1 V. Average CE of different electrolyte was also performed by Li‖Cu asymmetric cell based on the Aurbach method[40]. Polarization on Li‖Li symmetrical cell was conducted by pre-plated 5 mAh cm$^{-2}$ Li on Cu foil at the current density of 0.5 mA cm$^{-2}$ in a Li‖Cu asymmetric cell, followed by the stripping/plating at 1 mA cm$^{-2}$ for 1 h. Li‖NCM811 cells involving Li foil (20 μm) and NCM 811 laminate (8 mg cm$^{-2}$ or 20 mg cm$^{-2}$) by the N/P ratio of 2.5 or 1 were cycled at various specific currents in the voltage range of 4.4–2.8 V after formation cycles. The mass of the specific current and specific capacity refers to the mass of the active material in the positive electrode. Two or three cells were tested for a single electrochemical experiment. An additional piece of blank Al current collector (14 μm) and Al-coated (5 μm thick) stainless steel cathode case were used in all Li‖NCM811 cells to suppress electrolyte corrosion on stainless steel cathode case[8,14]. Above electrochemical measurements were performed by using LAND 2001A battery testing system. All electrochemical tests were carried out at ambient temperature without climatic chamber, except the ionic conductivity test by climatic chamber (DGBELL) at variable temperature with ±0.05 °C fluctuation.

**Reporting summary**

Further information on research design is available in the Nature Portfolio Reporting Summary linked to this article.

## Data availability

The data generated in this study have been deposited in the Zenodo database under accession code 7454646.

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

## Acknowledgements

A.C. acknowledges the support from the Swiss National Science Foundation (SNF) for funding of this research (200021-188572). J.W.C. acknowledges financial support from a National Research Foundation of Korea grant (NRF-2021R1A2B5B03001956, NRF-2018M1A2A2063340) and generous support from the Institute of Engineering Research (IOER), and Research Institute of Advanced Materials (RIAM) at Seoul National University.

## Author contributions

Y.Z., T.Z., and A.C. designed the concept. Y.Z. synthesized the molecule. Y.Z. and T.Z. tested electrochemical performance and characterized samples. M.M. conducted XPS measurements. Y.Z., T.Z., J.W.C., and A.C. wrote the paper. A.C. and J.W.C. supervised the research. All authors discussed the results and commented on the paper writing. Y.Z. and T.Z. contributed equally to this work.

## Competing interests

A.C., J.W.C., Y.Z., and T.Z. filed a patent application for the solvents and electrolytes described in this manuscript. Other authors declare no competing interests.
