## [Peer Review File · Nature Communications]

REVIEWER COMMENTS

Reviewer #1 (Remarks to the Author):

The author proposed a new-type ether-based solvent by rationally designing the substituent and substitution site, and therefore a moderate conductivity is preserved while increasing the oxidation stability. The electrochemical performances are very impressive both on the anode and full-cell level benefiting from the steric and electronic effect. The design concept is rational and very effective, and thus could be accepted after some minor revision as below:

1 The 2 M LiFSI-DMP electrolyte exhibits higher coulombic efficiency than the 2 M LiFSI-DME electrolyte in Li||Cu cells as shown in Figure 1c and 1d. However, the former degrades faster than the latter in the Li||Li symmetric cell, so what is the reason behind it?

2 The cell with 2 M LiFSI-DMP electrolyte seems to have higher capacity at the rate of 10 C than that with 2 M LiFSI-TFDMPA according to Figure 4d, which seems unreasonable. I wonder it is labeled by mistake or there are some other reasons?

3 Like this work, the rational molecular design for lithium salts, solvents and additives is a promising direction for constructing suitable electrolytes systems for LMBs. Therefore, some related work could be cited in the introduction part to highlight its importance. (For example, doi: 10.1021/acs.nanolett.2c01961)

4 There has been several work regarding the application of DME-based electrolyte in high-voltage cathode and should be cited. For example, the use of LiNO₃ has been proven to significantly widen the electrochemical window of DME-based electrolyte (doi: 10.1038/s41467-022-29761-z).

Reviewer #2 (Remarks to the Author):

See attached file below

This manuscript substituted the reactive proton on the ethyl moiety with a -CF₃ group to form fluorinated ether solvents 1,1,1-trifluoro-2,3-dimethoxypropane (TFDMP), which used for high voltage lithium metal batteries. By comparison with 1,2-dimethoxypropane (DMP), the electrochemical performance was systematically analyzed, which could present some important information to the readers of this journal. Therefore, this manuscript can be recommended for publication after the revisions are made for the listed comments and questions.

1. What's the reason that you focused on -CF₃, but not some other group?
2. Why did you choose 2M LiFSI as the lithium salt, instead of other lithium salts or other concentration?
3. Thermal stability is also an important index to evaluate the quality of electrolytes. What is the safety performance and thermal stability of the prepared electrolyte? It is recommended to supplement the SET test and thermal analysis.
4. It is recommended to supplement the ionic conductivity of the prepared electrolyte with different temperature changes.
5. The authors should list and compare the electrochemical performance of cells proposed herein with some other LMBs electrolytes proposed by other researchers.
6. The practical usage of 1,1,1-trifluoro-2,3-dimethoxypropane (TFDMP) in LMBs should be clarified. How do the authors look at the market prospect of the TFDMP?
7. Section "materials in methods", the details of TCI company should be provided.

Response to the reviewers' comments

Reviewer 1

The author proposed a new-type ether-based solvent by rationally designing the substituent and substitution site, and therefore a moderate conductivity is preserved while increasing the oxidation stability. The electrochemical performances are very impressive both on the anode and full-cell level benefiting from the steric and electronic effect. The design concept is rational and very effective, and thus could be accepted after some minor revision as below:

Response: We would like to thank the reviewer for his/her positive comment.

1. The 2 M LiFSI-DMP electrolyte exhibits higher coulombic efficiency than the 2 M LiFSI-DME electrolyte in Li||Cu cells as shown in Figure 1c and 1d. However, the former degrades faster than the latter in the Li||Li symmetric cell, so what is the reason behind it?

Response: Thank you for this question. The reason of shorter cycle life of 2 M LiFSI-DMP compared to 2 M LiFSI-DME in the Li||Li symmetric cell is the more serious reduction of DMP (LUMO: -0.20 eV) compared to DME (LUMO: -0.18 eV) as evidenced from the CV plots in the 1st scan in Supplementary Figure 7b, in which we observed a broad peak between 1.2-0.1 V for DMP. In addition, the XPS analysis of Li metal anode in Fig. 5f and 5g showed preferential reduction of DMP, proven by higher C-O peak ratio in the DMP-based electrolyte (24.7%) compared to the DME-based one (19.3%). While Coulombic efficiency reflects the Li plating reversibility, symmetrical cell analysis mainly shows interface stability. Similar phenomenon, that is higher CE but short lifetime of Li||Li symmetric cell, was also reported by Jiguang Zhang et al. (Energy Storage Materials, 2021, 34: 76-84).

Supplementary Figure 7. CV curves of Li|Cu half cells with 2 M LiFSI-DME (a), 2 M LiFSI-DMP (b) and 2 M LiFSI-TFDMP (c) electrolytes at a scan rate of 0.5 mV s^{-1} .

Fig. 5 C 1s profiles of SEI layers on Li anodes after 12 minutes sputtering with 2 M LiFSI-DME (f) and 2 M LiFSI-DMP (g).

2. The cell with 2 M LiFSI-DMP electrolyte seems to have higher capacity at the rate of 10 C than that with 2 M LiFSI-TFDMP according to Figure 4d, which seems unreasonable. I wonder it is labeled by mistake or there are some other reasons?

Response: Thank you for this comment. The lower capacity of 2 M LiFSI-TFDMP compared to 2 M LiFSI-DMP is mainly derived from the lower ionic conductivity of the TFDMP electrolyte. However, at 10 C rate, the full cell with 2 M LiFSI-TFDMP showed much higher CE of 99.4% than 2 M LiFSI-DMP, 95.0% by creating more stable interfaces on both sides.

3. Like this work, the rational molecular design for lithium salts, solvents and additives is a promising direction for constructing suitable electrolytes systems for LMBs. Therefore, some related work could be cited in the introduction part to highlight its importance. (For example, doi: 10.1021/acs.nanolett.2c01961)

Response: Thank you for the suggestion. This important work is cited in the introduction part.

On pg 2, “fluorinated amide (*Nano Letters*, 2022, 22(14): 5936-5943)”

4. There has been several work regarding the application of DME-based electrolyte in high-voltage cathode and should be cited. For example, the use of LiNO₃ has been proven to significantly widen the electrochemical window of DME-based electrolyte (doi: 10.1038/s41467-022-29761-z).

Response: Thank you for the suggestion. The referred work is now cited as below.

On pg 3, “Recently, Liu et al. reported anodic decomposition of DME was suppressed by electric double layer induced from LiNO₃ additive (*Nat Commun* 13, 2029 (2022)).”

Reviewer 2

This manuscript substituted the reactive proton on the ethyl moiety with a -CF₃ group to form fluorinated ether solvents 1,1,1-trifluoro-2,3-dimethoxypropane (TFDMP), which used for high voltage lithium metal batteries. By comparison with 1,2-dimethoxypropane (DMP), the electrochemical performance was systematically analyzed, which could present some important information to the readers of this journal. Therefore, this manuscript can be recommended for publication after the revisions are made for the listed comments and questions.

Response: We would like to thank the reviewer for his/her positive evaluation.

1. What's the reason that you focused on -CF₃, but not some other group?

Response: Thank you for this question. -CF₃ functional group can decrease the HOMO energy level to enhance the anodic stability of solvent and lower its solvation power to

promote the coordination between FSI anion and Li cation, which further enhances the anion decomposition to form the desired inorganic-rich SEI layer (Nat Commun 13, 2575 (2022)). In order to maintain the Li salt solvation ability, while imparting good oxidative stability, we resort to the –F groups that are distant from –O– atoms, as the direct attachment of –F group to the central alkyl moiety could substantially reduce the solvation ability of the solvent/diluent.

2. Why did you choose 2M LiFSI as the lithium salt, instead of other lithium salts or other concentration?

Response: Thank you for this comment. In the revised version, as shown in the Supplementary Figure 5, we systemically investigated the influence of lithium salts (LiPF₆, LiTFSI and LiFSI) and the salt concentrations (1M, 2M and 3M). As for the Li salts, LiFSI displayed much better CE performance than both LiPF₆ and LiTFSI. The salt concentration was also further compared, 1M salt showed the worst CE stability, whereas 2M and 3M exhibited similar and better CE stability, but higher salt concentration normally increases the cost and viscosity, accordingly, we used the 2M LiFSI.

Supplementary Figure 5. CE test of Li|Cu half cells using electrolytes with different salts (a) and different concentrations (b) at 3 mA cm⁻² with a cutoff capacity of 3 mAh cm⁻².

On pg 4, “The effect of different salts and salt concentrations in electrolytes was also compared (Supplementary Figure 5).”

3. Thermal stability is also an important index to evaluate the quality of electrolytes. What is the safety performance and thermal stability of the prepared electrolyte? It is recommended to supplement the SET test and thermal analysis.

Response: Thank you for this comment. Self-extinguishing time (SET) test and thermal analysis (TGA) data were added. TFDMP-based electrolyte showed the best safety performance evidenced by the lowest SET value owing to the presence of the fluorine atoms in the solvent (Nature Nanotechnology, 2018, 13(8), 715-722). The electrolytes showed similar thermal stability below 200 °C (Supplementary Figure 13). Notably, only TFDMP-based electrolyte maintained thermal stability from 200 to 300 °C.

Supplementary Table 3. Self-extinguishing time (SET) test of different electrolytes.

	2 M LiFSI-DME	2 M LiFSI-DMP	2 M LiFSI-TFDMP
SET (s g ⁻¹)	83	86	75

Supplementary Figure 13. TGA analysis of different electrolytes.

On pg 6, “Safety performance and thermal stability of each electrolyte were also evaluated by self-extinguishing time (SET) and thermogravimetric analysis (TGA) (Supplementary Table 3 and Figure 13).”

On pg 16, Materials characterizations, “TGA analysis of electrolytes was conducted using a TGA/DSC 3+ STAR^e System (Mettler Toledo) under N₂ gas flow (5 mL/min) at the ramp rate of 10 °C/min from 25 °C to 600 °C using a 100 µL aluminum crucible with a predrilled lid.”

4. It is recommended to supplement the ionic conductivity of the prepared electrolyte with different temperature changes.

Response: Thank you for this suggestion, ionic conductivity of electrolytes at different temperatures was added (Supplementary Figure 11).

Supplementary Figure 11. Ionic conductivity of electrolytes at different temperatures.

5. The authors should list and compare the electrochemical performance of cells proposed herein with some other LMBs electrolytes proposed by other researchers.

Response: Thank you for this comment. Electrochemical performance comparison of LMBs electrolytes was added in Supplementary Table 4. Performance of LMBs using 2 M LiFSI-TFDMP is competitive in terms of rate capability and N/P ratio.

Supplementary Table 4. Comparison of electrochemical performance of full cells with different electrolytes

Reference	Electrolyte	Cathode	Loading	Electrochemical performance
This work	2 M LiFSI-TFDMP	NCM811	20 mg cm ⁻²	0.5C/0.5C, N/P=1, 200 cycles (81%) 1C/1C, N/P=1, 145 cycles (88%)
Nature Energy, 2022, 7(1): 94-106	1.2 M LiFSI/F5DE E	NCM811	4.9 mAh cm ⁻²	0.2C/0.3C, N/P=2, 200 cycles (80%)

Nature Energy , 2020, 5(7): 526-533	1 M LiFSI/FDMB	NCM811 NCM523	2 mAh cm ⁻² 2 mAh cm ⁻²	0.33C/0.33C, N/P=2, 110 cycles (~85%) 0.33C/0.33C, N/P=2.5, 210 cycles (100%)
Nature communications , 2022, 13, 2029	LiFSI- LiNO ₃ /DME	NCM811	20.3 mg cm ⁻²	2C/2C, N/P=2.31, 200 cycles (82%)
Nature communications , 2022, 13, 5431	1.3 M LDC	NCM811	21 mg cm ⁻²	0.33C/1C, N/P=1, 200 cycles (80%)
Nature communications , 2022, 13, 2575	2 M LiFSI- DTDl	NCM811	5 mg cm ⁻²	0.5C/0.5C, N/P=4, 200 cycles (84%)
Energy & Environmental Science , 10.1039/D2EE01756C	1.5 M LiFSI DMMS	NCM811 LCO	2.3 mAh cm ⁻² 3 mAh cm ⁻²	0.33C/0.66C, N/P=1.7, 350 cycles (80%) 0.33C/0.66C, N/P=1.3, 200 cycles (95%)
ACS Applied Energy Materials , 2022, 5(6): 7784-7790	1.5 M LiFSI- 8TTD-2DME	NCM811	8 mg cm ⁻² 20 mg cm ⁻²	0.5C/0.5C, N/P=2.5, 160 cycles (75%) 0.3C/0.3C, N/P=1, 100 cycles (69%)

On pg 11, “Performance comparison of full cells with different electrolytes was summarized in Supplementary Table 4, LMBs using 2 M LiFSI-TFDMP are quite competitive.”

6. The practical usage of 1,1,1-trifluoro-2,3-dimethoxypropane (TFDMP) in LMBs should be clarified. How do the authors look at the market prospect of the TFDMP?

Response: Thank you for this question. From the practical usage standpoint, TFDMP can be synthesized by 100 grams scale in the lab. As fluorinated compounds have already been used in commercial cells, we don't see any critical issue related to the scalability of TFDMP although an in-depth analysis is needed for more accurate cost evaluation.

7. Section “materials in methods”, the details of TCI company should be provided.

Response: Thank you for your suggestion. The company information was added in

materials section as below.

On pg 15. “1,2-dimethoxypropane (DMP) was purchased from Tokyo Chemical Industry (TCI).”

REVIEWERS' COMMENTS

Reviewer #1 (Remarks to the Author):

The authors has addressed the concerns and can be accepted in the present form.

Reviewer #2 (Remarks to the Author):

This manuscript has been well-revised. I have no more comments.